# Insights into the Frequency and Distinguishing Features of Sleep Disorders in Pediatric Palliative Care Incorporating a Systematic Sleep Protocol

**DOI:** 10.3390/children8010054

**Published:** 2021-01-17

**Authors:** Larissa Alice Dreier, Boris Zernikow, Kathrin Stening, Julia Wager

**Affiliations:** 1PedScience Research Institute, 45711 Datteln, Germany; b.zernikow@kinderklinik-datteln.de (B.Z.); j.wager@deutsches-kinderschmerzzentrum.de (J.W.); 2Department of Children’s Pain Therapy and Paediatric Palliative Care, Faculty of Health, School of Medicine, Witten/Herdecke University, 58448 Witten, Germany; 3Paediatric Palliative Care Centre, Children’s and Adolescents’ Hospital, 45711 Datteln, Germany; k.stening@kinderpalliativzentrum.de

**Keywords:** sleep, palliative care, pediatrics, neurobehavioral manifestations

## Abstract

Currently, no concrete figures on sleep disorders and sleep characteristics in children and adolescents with life-limiting conditions (LLC) and severe neurological impairment (SNI) based on pediatric palliative care professionals’ assessment and following an official classification system such as the International Classification of Sleep Disorders (ICSD-3) exist. The ICSD-3 sleep disorders of inpatient children and adolescents with LLC and SNI (*N* = 70) were assessed by professionals using a recording sheet (two-year recruitment period). A systematic sleep protocol was applied to identify patients’ sleep characteristics. Of these patients, 45.6% had sleep disorders, with the majority of them experiencing two different ones. Overall, the most frequently identified disorders were Chronic Insomnia and Circadian Sleep–Wake Disorder. Patients experiencing Chronic Insomnia showed more sleep phases during the daytime and more waking phases at nighttime than those unaffected. Patients with and without a Circadian Sleep–Wake Disorder additionally differed in the length of sleep phases during the daytime. Rapid changes between wakefulness and sleep were specifically characteristic of Hypersomnia. The study provides important insights into the prevalence and characteristics of individual ICSD-3 sleep disorders in pediatric palliative care. The findings may contribute to a targeted and efficient diagnosis and therapy of distressing sleep problems in seriously ill patients.

## 1. Introduction

Every year, around 21 million children and adolescents worldwide are affected by life-limiting conditions (LLC) requiring palliative care [1]. These conditions are mainly genetic, neurological, or metabolic and, less commonly, oncological diseases [2]. A common feature these children and adolescents share is a variety of stressful symptoms [3,4]. In this regard, a particularly challenging aspect is that many of the affected individuals are non-verbal due to severe neurological impairment (SNI) and are therefore unable to communicate directly [3,5]. With a prevalence rate between 50 and 80%, sleep problems represent one of the most common symptoms in the population [6,7,8,9,10]. Sleep disorders do not only place an enormous burden on the sick child or adolescent but also on its parents, who themselves consequently experience a variety of physical and psychological problems [6,11,12,13,14].

The term “sleep disorders” constitutes a collective name for a wide range of disparate single diseases, which according to the International Classification of Sleep Disorders (ICSD-3) [15] can be divided into the main categories of Insomnia, Sleep-Related Breathing Disorders, Central Disorders of Hypersomnolence, Circadian Rhythm Sleep–Wake Disorders, Parasomnias, and Sleep-Related Movement Disorders [15]. Identifying symptoms in general and the manifold sleep disorders in particular with as much accuracy as possible poses one of the greatest challenges for pediatric palliative care service providers, who try to ensure that children and families are offered appropriate therapeutic approaches with the aim of establishing an acceptable level of comfort [2,3,5,6,12,16].

Today, a variety of subjective measures such as sleep questionnaires or sleep diaries and objective measures such as systematic observation tools or Polysomnography are available for sleep diagnostics [6,16,17,18,19,20].

Based on information mainly gained from sleep questionnaires filled out by parents, assumptions on the frequency of sleep problems in children and adolescents with LLC and SNI can be made: the severely ill seem to show symptoms of the whole spectrum of ICSD sleep disorders, whereby the most common ones reported include problems with falling and staying asleep, symptoms relating to the circadian sleep-wake rhythm and symptoms related to nocturnal respiration [8,13,21,22,23]. Although this parent information is essential, the official billable diagnosis has ultimately to be made by professional caregivers. Interestingly, to the best of our knowledge, no concrete figures are available on the frequency and distribution with which these professionals assign what sleep diagnoses to children and adolescents with LLC and SNI. Consequently, it is also unclear to what extent affected and unaffected individuals differ regarding objective sleep parameters (e.g., duration of sleep and waking phases), that is, whether specific sleep diagnoses are associated with typical sleep-related characteristics in seriously ill children and adolescents.

The present study will therefore evaluate (a) which ICSD sleep disorders occur how frequently and in which combination in the population of children and adolescents with LLC and SNI and (b) whether characteristic features can be identified for children and adolescents with and without specific sleep disorders with regard to objective sleep parameters.

## 2. Materials and Methods

### 2.1. Participants

Data were prospectively collected over a two-year period (January 2017–December 2018) on an inpatient pediatric palliative care unit. The pediatric palliative care unit is affiliated with a tertiary care children’s hospital and provides highly specialized pediatric palliative care for annually approximately *N* = 170 patients with mainly neurological underlying diseases and their families [24]. Opened in 2010, it was the first pediatric palliative care unit in Europe. A total of eight beds in single-patient rooms and a qualified multiprofessional team (e.g., consisting of physicians, nurses, and various therapists) that pursues a holistic bio-psycho-social-spiritual care approach were available.

Children and adolescents who met the following criteria were included: (a) LLC, (b) SNI, (c) age between 1–25 years, (d) inpatient stay of at least 7 days.

Families in acute crisis, children/adolescents whose life expectancy would most probably be less than four weeks, and patients whose parents refused to participate in the study were excluded. Since the data collected were intended to answer further questions beyond those of this study, one additional exclusion criterion (insufficient language skills of parents) needed to be defined during the study conduct. Ethical approval was obtained by the Ethics Committee of the Children’s and Adolescents’ Hospital Datteln (approval code: 2017/03/08/BZ1); all families provided informed consent to participate in the study.

### 2.2. Assessment of Sleep Disorders

To systematically record the professional caregivers’ judgment on the presence or absence of a child’s or adolescent’s sleep disorder(s), a recording sheet based on the ICSD-3 main categories was drawn up by the research team in close cooperation with an internal team of pediatric palliative care practitioners. For each diagnosis, this recording sheet included all ICSD-3 criteria for the corresponding diagnosis. Sleep-Related Breathing Disorders were not queried since the definitive diagnosis of these usually requires a Polysomnography and could therefore not have been reliably assessed in the context of this study [25].

ICSD criteria for those sub-diagnoses that explicitly consider the presence of an underlying disease were listed where possible. The final version of the recording sheet listed the diagnoses Chronic Insomnia Disorder (Chronic Insomnia in the following), Hypersomnia Due to a Medical Disorder (Hypersomnia in the following), Parasomnia Due to a Medical Condition (Parasomnia in the following), Sleep-Related Movement Disorder Due to a Medical Disorder (Sleep-Related Movement Disorder in the following), and Circadian Sleep–Wake Disorder Not Otherwise Specified (Circadian Sleep-Wake Disorder in the following).

### 2.3. 24-Hours Sleep Protocol

The 24-h sleep protocol was developed at the Paediatric Palliative Care Centre in Datteln, Germany, to enable a simple yet systematic and objective sleep and wakefulness documentation of all admitted patients. In the protocol, sleep and wake phases during the day and the night can be entered in a 15-min grid and marked in different, predefined colors. Additionally, the events “pain”, “spasticity”, “seizures”, “restlessness”, “screaming”, “dyspnea”, “vomiting/constipation” and “receiving meals” can be documented using standardized abbreviations.

For the study purpose, “quick phase change” was defined as an additional event. It describes a quick change between sleep and wake phases for which no concrete number/duration could be determined (“hatching” in the protocol).

“Daytime” was defined as the period from 8 am to 8 pm, “nighttime” from 8 pm to 8 am. Hereinafter, the term “day” is used to refer to a full 24-h day. The first day was determined as the day on which a child/adolescent spent a full 24 h on the unit (usually the day after the admission day). Days 1–4 were defined and grouped as the acclimatization period and days 5–7 as reference days. Data of the acclimatization period were not analyzed since children or adolescents admitted to a hospital often show no picture or a distorted picture of their symptoms in the first three to four days [26,27].

### 2.4. Data Collection

One week after the admission of a patient, a team consisting of a variable constellation of nurses and physicians assessed the presence of sleep disorders using the recording sheet: the criteria of each individual ICSD sleep disorders of the recording sheet were evaluated, and the professionals’ assessment was documented by a member of the study team. The professionals’ judgment was based on the child’s/adolescent’s data gathered during the first week on the unit and on their own clinical expertise. The research team extracted the patient’s demographic data from the patient file and reviewed the corresponding 24-h sleep protocol.

### 2.5. Data Analysis

Based on the professionals’ entries on the recording sheet, the distribution and frequency of all five sleep disorders were analyzed using descriptive statistics. Here, a distinction is made between the age groups “children” (defined age range: 1–12 years) and “adolescents” (defined age range: 13–23 years). “Children” are further subdivided into “toddlers” (defined age range: 1–3 years) and “older children” (defined age range: 4–12 years). For identifying patients’ sleep-related characteristics, their 24-h sleep protocols were then descriptively evaluated with respect to the following parameters:Core phases: Sleep and wake phases during the daytime and nighttime (number and duration)Parameters of phase initiation: Sleep onset time (start of first sleep phase during the nighttime) and awakening time (start of first wake phase during the daytime)Events in the core phases during the daytime and nighttime (number, as duration could not be reliably interpreted due to the way of standard recording).

For Parasomnia and Sleep-Related Movement Disorder, 24-h sleep protocol data are not reported since their core features are not characterized by abnormalities in the sleep or wake phases and therefore cannot be assessed with such an instrument.

Spearman-Rho correlation analyses were performed to provide exploratory evidence of parameters in which children and adolescents with and without attributed sleep disorders may differ. With a simultaneous significant correlation of parameters that are conditionally related to each other (e.g., number of sleep and waking phases during daytime/nighttime), only those parameters indicating the pathological behavior typical of a disorder were considered. Group differences were determined using Mann–Whitney U tests (corrected for Bonferroni-Holm). All data were analyzed using the SPSS statistics software (IBM, version 26).

## 3. Results

### 3.1. Sample Characteristics

During the study period, *N* = 70 patients (female: *n* = 44, 62.9%; male: *n* = 26, 37.1%) with an age between 1 and 23 years (M = 9.0, SD = 6.5) were included. *n* = 46 (65.7%) were children and *n* = 24 (34.3%) were adolescents. When further subdivided, *n* = 20 (28.6%) of the children were toddlers and *n* = 26 (37.1%) were older children. Almost 60% (59.9%) of the patients had the long-term care grade V, which is the highest possible in Germany and which expresses severe impairments of autonomy and skills [28]. Overall, these children and adolescents showed a broad range of different underlying neurological, metabolic, and muscular diseases (Table 1). The three most frequently represented categories with ICD-10 codes P91, E70–E90, and Q87, for example, included underlying diseases such as hypoxic-ischemic encephalopathy, neuronal ceroid lipofuscinosis, Krabbe disease, and different trisomies. Reflecting the defined inclusion criteria, all of these underlying diseases were associated with SNI (e.g., impaired verbal communication skills, global developmental delays).

### 3.2. Sleep Disorders Ascribed by Professionals

Overall, *n* = 32 (45.7%) patients had a sleep disorder, while *n* = 38 (54.3%) did not. In both the “toddler” subgroup and the “older children” subgroup, exactly half of the patients had a sleep disorder (toddler: *n* = 10; older children: *n* = 13). In the “adolescents” subgroup, *n* = 9 (37.5%) patients had a sleep disorder and *n* = 15 (62.5%) did not. Of the total sample, *n* = 4 (12.5%) patients fulfilled all criteria for one, *n* = 15 (46.9%) for two, *n* = 12 (37.5%) for three, and *n* = 1 patient (3.1%) for four different sleep disorders. The most common sleep disorders were Chronic Insomnia (*n* = 30, 93.8%), followed by Circadian Sleep–Wake Disorder (*n* = 27, 84.4%), Hypersomnia (*n* = 10, 31.3%), Sleep-Related Movement Disorder (*n* = 4, 12.5%), and Parasomnia (*n* = 3, 9.4%).

In the subgroups, *n* = 10 (50%) toddlers, *n* = 11 (42.3%) older children, and *n* = 9 (37.5%) adolescents experienced Chronic Insomnia; *n* = 8 (40%) toddlers, *n* = 11 (42.3%) older children, and *n* = 8 (33.3%) adolescents experienced a Circadian Sleep–Wake Disorder; *n* = 1 (5%) toddler, *n* = 4 (15.4%) older children, and *n* = 5 (20.8%) adolescents showed Hypersomnia; *n* = 2 (10%) toddlers, *n* = 2 (7.7%) older children, and none of the adolescents experienced a Sleep-Related Movement Disorder; *n* = 1 (5%) toddler, *n* = 2 (7.7%) older children, and none of the adolescents showed Parasomnia” Chi-square tests revealed that there was no difference in prevalence of any sleep disorder across patient age subgroups (all *p* > 0.05).

In the overall group, the most frequent combination of sleep disorders was Chronic Insomnia plus Circadian Sleep–Wake Disorder, and the second most frequent combination was Chronic Insomnia plus Hypersomnia plus Circadian Sleep–Wake Disorder (Table 2).

### 3.3. Analyses of the 24-Hours Sleep Protocols

Correlation analyses were shown for Chronic Insomnia (number) and Circadian Sleep–Wake Disorder (number, duration) to be significantly related to the core phases of the 24-h sleep protocol, whereas for Hypersomnia, they were only related to the quick phase change event. A Circadian Sleep–Wake Disorder likewise correlated with the quick phase change event. No sleep disorder was associated with the phase initiation parameters (sleep onset, awakening time; Table 3).

Figure 1 shows the differences between patients with and without the different sleep disorders with respect to the relevant pathological behaviors (derived from ICSD-3 criteria): patients with Chronic Insomnia showed an average number of M = 1.5 sleep phases (SD = 1.06) during the daytime (patients unaffected: M = 0.91, SD = 0.78) and an average number of M = 2.2 wake phases (SD = 1.1) during the nighttime (patients unaffected: M = 1.61, SD = 0.79). A Mann–Whitney-U test with the grouped variable (days 5–7) confirmed that these differences were statistically significant (number of sleep phases during the daytime: U = −2.53, *p* = 0.01; number of wake phases during the nighttime: U = −2.55; *p* = 0.01).

Patients with Hypersomnia showed the quick phase change event more frequently during the daytime (M = 0.33, SD = 0.38) than those without the corresponding disorder (M = 0.10, SD = 0.19; U = −2.0, *p* = 0.04).

An attributed Circadian Sleep–Wake Disorder was associated with significantly more wake phases at nighttime (patients affected: M = 2.23, SD = 1.16; patients unaffected: M = 1.66, SD = 0.79; U = −2.0, *p* = 0.03) and with more and longer sleep phases during the daytime (number: patients affected: M = 1.65, SD = 1.05, patients unaffected: M = 0.86, SD = 0.74, U = −3.27, *p* < 0.01; duration: patients affected: M = 2 h, 15 min, SD = 1 h, 43 min, patients unaffected: M = 1 h, 16 min, SD = 1 h, 9 min, U = −2.52, *p* = 0.01). The number of quick phase change events during the daytime revealed only descriptively a difference between patients with and without Chronic Sleep–Wake Disorder (*p* > 0.05 after Bonferroni–Holm correction).

## 4. Discussion

This study aimed to provide concrete numbers for the frequency and combination of specific ICSD-3 sleep disorders in a highly vulnerable population. Characteristic sleep-related patterns among affected and unaffected children and adolescents were to be identified.

About half of the children and adolescents included in this study had sleep disorders; most of them were diagnosed with more than one sleep disorder. The high prevalence rate of sleep disorders corresponds to related work, which indicates that, firstly, children and adolescents with SNI are apparently more frequently affected by sleep disorders than children and adolescents with a different type of underlying condition and, secondly, that different types of sleep disorders usually tend to occur cumulatively [8,29,30,31,32,33].

Chronic Insomnia and Circadian Sleep–Wake Disorder were the most prevalent sleep disorders in our focused sample. This finding is in line with results from related studies based on parental judgment [13,22,34,35,36]. By consulting professionals on the basis of official ICSD-3 criteria, we further extend these statements regarding the prevalence of sleep disorders in pediatric palliative care patients, with the so far underrepresented diagnostic assessment of an essential group of persons.

Interestingly, Chronic Insomnia and Circadian Sleep–Wake Disorder were often linked; specifically, professionals usually attributed both sleep disorders to patients. Different reasons are conceivable for this: since the presence of Chronic Insomnia according to ICSD-3 is one of the features of a Circadian Sleep–Wake Disorder, it is possible that in these patients to whom both diagnoses were ascribed, the professional caregivers considered a Circadian Sleep–Wake Disorder as the core diagnosis, and the assigned Chronic Insomnia was merely intended to express the fulfilled ICSD-3 criterion. Another explanation could be that the professionals intended to underline that the problematic (sleep) behavior was particularly evident at night and the daytime behavior had changed accordingly, assuming that a shift in for example the time of falling asleep already implies a disturbed sleep–wake rhythm.

The same reasoning may apply to the second most frequent combination of sleep disorders in this study, in which Hypersomnia occurred in addition to the two previously mentioned disorders. Possibly, this was to indicate that the child’s or adolescent’s behavior at night and during the day was comparably conspicuous. The Circadian Sleep–Wake Disorder could again have been assigned as a “resulting” disturbance here.

Since the aim of this study was not to ascertain the correctness of the professionals’ judgment, the above-mentioned considerations cannot be conclusively clarified at this point. Upcoming research efforts should evaluate whether the parallel occurrence of specific sleep disorders in severely ill patients can be replicated. This is important since statements on the prevalence of sleep disorders are often solely analyzed and reported across an overall group, making conclusions about frequent combinations of sleep disorders on an individual level hardly possible.

By analyzing the 24-h sleep protocols, we aimed to describe typical characteristics of patients with LLC and SNI with and without Chronic Insomnia, Hypersomnia, and Circadian Sleep–Wake Disorder.

Obviously, patients from our sample experiencing Chronic Insomnia can be distinguished from unaffected ones by both a higher number of nocturnal wake phases and sleep phases during the daytime. Additionally, a large agreement was identified between the typical characteristics of Chronic Insomnia and Circadian Sleep–Wake Disorder, which can possibly be explained by the considerations already mentioned. Interestingly, we could not find any association between the professional caregivers’ diagnosis and the time of falling asleep and waking up. Sleep or wakefulness behavior that is untypical for the time of day or night thus seems to be more decisive than the question of how long this untypical behavior will last or behaviors that additionally emerge during this time. The pathological behavior’s length and the events would possibly become more relevant under the question of which therapeutic approach (e.g., which medication) is to be chosen under which hypothesis (e.g., sleep disorder due to nighttime seizures). Nonetheless, it should be taken into account that we defined the beginning and end of the daytime and nighttime somewhat artificially. Perhaps the professionals were more oriented towards times that were geared to the individual patient or other parameters that could not be captured by our rigidly defined time boundaries.

Hypersomnia seems to be characterized by a rapid, hardly quantifiable change between wakefulness and sleep during the day. Some evidence suggests that the establishment and maintenance of an adequate arousal level may be impaired, particularly in the case of neurological diseases accompanied by functional or structural changes of the brain and central nervous system [37,38]. Since the majority of the participating patients had neurological diseases, it is plausible that these processes might have been disturbed. The quick phase change event during the day could therefore have been the expression of two ambivalent processes: an intense craving for sleep and disturbed arousal processes, which perhaps interrupted the initiation of sleep. Further research, incorporating current neurochemical knowledge, is necessary to gain better insight into this phenomenon.

### Limitations of the Study

The major limitation of our study is that due to the chosen instrument and the methodological approach, we can only make statements for a selection of pediatric sleep disorders. Because some important other sleep disorders (Sleep-Related Breathing Disorders; Parasomnia and Sleep-Related Movement Disorders from identifying sleep-related characteristics) were not considered in this study, it is vital that this is taken into account when interpreting the study results and should not be misconstrued to mean that these non-focused sleep disorders are not present or not relevant in the sample studied. To enable valid statements regarding typical sleep-related characteristics of patients with sleep disorders, it is important that a wider range of disorders is investigated in future studies.

Another point to be taken up in this context is the sample size of *N* = 70 patients. Although this was sufficient to answer the study’s key questions, we cannot claim a comprehensive coverage of all patients with LLC and SNI. The multitude of exclusion reasons for parents and children/adolescents identified once again shows the special challenge of the setting of pediatric palliative care. It will be impossible to abolish them, but in the research context, we must create access points allowing us to reach the greatest possible number of affected individuals with as little strain as possible. The use of resource-efficient measures such as the 24-h sleep protocol can contribute to this since the hurdle of study participation is likely to increase proportionately with a reduction in the associated effort.

Regarding the diagnosis of sleep disorders in this study, another important limitation must be mentioned: the judgment about the presence or absence of a corresponding sleep disorder was based only on the professionals’ subjective judgment and was not supplemented by additional (objective) measures.

Moreover, we cannot make any statements about the decision-making principles and processes used by professional pediatric palliative care providers when assigning a sleep diagnosis, which is important for gaining better insights into the diagnostic process and avoiding possibly wrong therapeutic decisions. In addition, further research could clarify whether official diagnostic systems and the criteria listed therein are appropriate for determining sleep disorders in the highly specific population of children and adolescents with LLC and SNI or if the long-term efforts to develop sensibly adapted standards should be pursued.

## 5. Conclusions

This study gave an insight into the diagnosis of certain sleep disorders made by professional caregivers in inpatient pediatric palliative care, showed that the 24-h sleep protocol is a promising objective tool for use in this unique setting, and revealed typical characteristics of patients with and without sleep disorders of three ICSD-3 major groups. Areas of future interesting research have been mentioned above. In general, future research efforts should build on our study’s findings in order to continuously improve the quality of the diagnosis of sleep disorders in seriously ill children and adolescents.

## Figures and Tables

**Figure 1 children-08-00054-f001:**
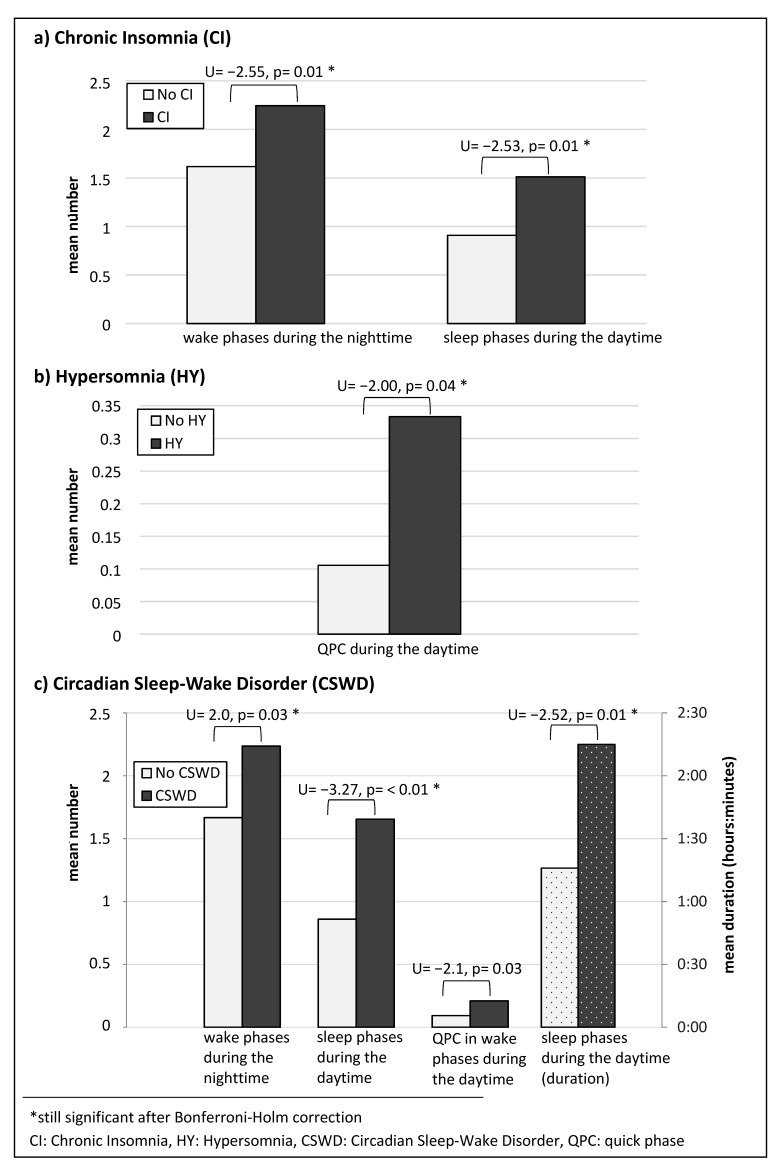
Mean values and differences in the number (no pattern) and duration (dotted) of the identified sleep/wake parameters between patients with and without Chronic Insomnia (**a**), Hypersomnia (**b**), and Circadian Sleep–Wake Disorder (**c**).

**Table 1 children-08-00054-t001:** Frequency of individual sleep disorders for patients with different underlying diseases (classification based on ICD-10).

ICD-10 Code—Group (*n*,% in Total Sample) Sleep Disorders According to ICSD-3 ^a^ (*n*,% within Underlying Disease)				
Chronic Insomnia	Hypersomnia	Parasomnia	Sleep-Related Movement Disorder	Circadian Sleep–Wake Disorder
**G12**—Spinal muscular atrophy and related syndromes (*n* = 3, 4.3)				
1 (33.3)	-	-	-	1 (33.3)
**G31**—Other degenerative diseases of nervous system, not elsewhere classified (*n* = 3, 4.3)				
3 (100)	1 (33.3)	-	1 (33.3)	2 (66.7)
**G71**—Primary disorders of muscles (*n* = 2, 2.9)				
-	-	-	-	1 (50.0)
**G93.1**—Anoxic brain damage, not elsewhere classified (*n* = 7, 10.0)				
2 (28.6)	1 (14.3)	1 (14.3)	1 (14.3)	1 (14.3)
**Q04.8**—Other specified congenital malformations of brain (*n* = 4, 5.7)				
1 (25.0)	1 (25.0)	-	-	1 (25.0)
**Q04.9**—Congenital malformation of brain, unspecified (*n* = 5, 7.1)				
3 (60.0)	2 (40.0)	-	-	2 (40.0)
**Q93.8**—Other specified disorders of brain (*n* = 3, 4.3)				
2 (66.7)	1 (33.3)	-	-	2 (66.7)
**E70–E90**—Metabolic disorders (*n* = 11, 15.7)				
6 (54.5)	1 (9.1)	-	-	6 (54.5)
**Q87**—Other specified congenital malformation syndromes affecting multiple systems (*n* = 10, 14.3)				
6 (60.0)	3 (30.0)	-	-	5 (50.0)
**Q89**—Other congenital malformations, not elsewhere classified (*n* = 2, 2.9)				
1 (50.0)	-	-	-	1 (50.0)
**Q93.9**—Disorders of brain, unspecified (*n* = 2, 2.9)				
-	-	-	-	-
**P91**—Other disturbances of cerebral status of newborn (*n* = 18, 25.7)				
5 (27.8)	-	2 (11.1)	2 (11.1)	5 (27.8)

^a^ multiple sleep disorders per patient possible.

**Table 2 children-08-00054-t002:** Frequencies of assigned sleep disorders (vertical), as well as type and frequency of their combinations (horizontal) in patients to whom more than one disorder was ascribed (grey boxes indicate a sleep disorder).

	Type *n* (%)					
Number of Assigned Sleep Disorders	Chronic Insomnia	Hypersomnia	Parasomnia	Sleep-Related Movement Disorder	Circadian Sleep–Wake Disorder	*n* (%)
1						3 (9.4)
						1 (3.1)
2						14 (43.8)
						1 (3.1)
3						9 (28.1)
						2 (6.3)
						1 (3.1)
4						1 (3.1)
	30 (93.8)	10 (31.3)	3 (9.4)	4 (12.5)	27 (84.4)	

**Table 3 children-08-00054-t003:** Correlations between patients having different sleep disorders and (a) the core phases (sleep, wakefulness), (b) the phase initiation parameters, and (c) the events within the core phases. The pathological behavior typical of a specific sleep disorder (derived from ICSD-3 criteria) is highlighted in bold letters.

		Nighttime				Daytime			
core phases	sleep disorder	sleeping phase		waking phase		sleeping phase		waking phase	
		nr	du	nr	du	nr	du	nr	du
	Chronic Insomnia	0.32 **	−0.05	**0.30 ****	0.15	**0.30 ***	0.19	0.32 **	0.03
	Hypersomnia	0.19	−0.01	0.16	0.06	0.15	0.19	0.11	−0.14
	Circadian Sleep–Wake Disorder	0.28 *	−0.19	**0.25 ***	0.20	**0.39 ****	**0.30 ***	0.35 **	−0.14
phase initiation	sleep disorder	sleep onset time				awakening time			
		r				r			
	Chronic Insomnia	−0.12				−0.09			
	Hypersomnia	−0.07				−0.00			
	Circadian Sleep–Wake Disorder	−0.09				0.03			
		nighttime				daytime			
Events ^a^	sleep disorder	sleeping phase		waking phase		sleeping phase		waking phase	
		type	r	type	r	type	r	type	r
	Chronic Insomnia								
	Hypersomnia	QPC	0.24 *					QPC	**0.24 ***
	Circadian Sleep–Wake Disorder							QPC	**0.25 ***

nr: number of phases, du: duration in hours and minutes, QPC: quick phase change. ^a^ due to the multitude of possible events, only those for which significant correlations could be identified are listed (tested, but not significant unless otherwise stated above: pain, spasticity, seizures, restlessness, screaming, dyspnoea, vomiting/obscuration, meals, QPC). * *p* < 0.05, ** *p* < 0.01.

## Data Availability

The data presented in this study are available on request from the corresponding author.

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
