# Peer review of "Insights into the Frequency and Distinguishing Features of Sleep Disorders in Pediatric Palliative Care Incorporating a Systematic Sleep Protocol"

_children, 2021, doi:10.3390/children8010054_

Round 1

Reviewer 1 Report

This study describes sleep disorders and sleep characteristics in 70 children and adolescents with life-limiting conditions (LLC) and severe neurological impairment (SNI). Professionals assessed the presence od sleep disorders on the basis of the ICSD-3.

About half of the children and adolescents included in this study had sleep disorders and the most frequently identified disorders were “Chronic Insomnia” and “Circadian Sleep-Wake Disorder”.

This is essentially a decriptive study that confirms previous findings.

The main limitations of the study is represneted by the "diagnosis". Appropriately, the authors did not include "Sleep Related Breathing Disorders”, due to the lack of objective sleep measures.

Unfortunately, this limitation also regards some "Parasomias" and some "Sleep Related Movement Disorders" that were actually considered (i.e., "diagnosed"). This represents a basic limitation of the study and it should reasonably lead to an undertimate of the prevalence of these disorders.

Another basic limitation is represented by the wide age range (1-23 years). The prevalence of some specific sleep disorders widely changes agross these ages.

Concerning the first point, the authors can only mention and discuss this basic limitation. Furthermore, they have to describe how professianls "diagnosed" these classes of disorders.

Concerning the second point, the authors have to report data as a function of some different age groups. This is mandatory

Author Response

Insights into the frequency and distinguishing features of sleep disorders in pediatric palliative care incorporating a systematic sleep protocol

children-1049545

This study describes sleep disorders and sleep characteristics in 70 children and adolescents with life-limiting conditions (LLC) and severe neurological impairment (SNI). Professionals assessed the presence od sleep disorders on the basis of the ICSD-3.

About half of the children and adolescents included in this study had sleep disorders and the most frequently identified disorders were “Chronic Insomnia” and “Circadian Sleep-Wake Disorder”.

This is essentially a descriptive study that confirms previous findings.

The main limitations of the study is represented by the "diagnosis". Appropriately, the authors did not include "Sleep Related Breathing Disorders”, due to the lack of objective sleep measures.

Unfortunately, this limitation also regards some "Parasomias" and some "Sleep Related Movement Disorders" that were actually considered (i.e., "diagnosed"). This represents a basic limitation of the study and it should reasonably lead to an underestimate of the prevalence of these disorders.

Another basic limitation is represented by the wide age range (1-23 years). The prevalence of some specific sleep disorders widely changes across these ages.

Concerning the first point, the authors can only mention and discuss this basic limitation. Furthermore, they have to describe how professionals "diagnosed" these classes of disorders.

[…]

Thank you very much for your comments and the opportunity to revise our paper with regard to your suggestions. Regarding your first named aspect, we have further elaborated on the basic limitation in the discussion to emphasize it more clearly (p. 9, ll. 296-301;Because some important other sleep disorders (Sleep-Related Breathing Disorder;  Parasomnia and Sleep-Related Movement Disorders when identifying sleep-related characteristics) were not considered in this study, it is vital that this is taken into account when interpreting the study results and should not be misconstrued to mean that these non-focused sleep disorders are not present or not relevant in the sample studied.”).

In the discussion, we also included a passage on the methodology of diagnosing sleep disorders in this study as a relevant limitation (p. 9, ll. 311-314; Regarding the diagnosis of sleep disorders in this study, another important limitation must be mentioned: The judgment about the presence or absence of a corresponding sleep disorder was based only on the professionals' subjective judgment and was not supplemented by additional (objective) measures.”).

We have included two sentences in the methods that further clarify how professionals diagnosed the ICSD sleep disorders (p. 2, ll. 90-91;For each diagnosis, this recording sheet included all ICSD-3 criteria for the corresponding diagnosis.” and p. 3, ll. 119-121; “[…] the criteria of each individual ICSD sleep disorders of the recording sheet were evaluated, and the professionals' assessment was documented by a member of the study team.”).

We have made minor adjustments to the abstract to further optimize understanding of the methodology (“daytime” instead of day, “nighttime” instead of night).

[…]

Concerning the second point, the authors have to report data as a function of some different age groups. This is mandatory.

Thank you very much for this comment. To comply with your request, we divided our sample into different age groups. Corresponding additions were made at various points in the results:

  • 3, ll. 127-130;Here, a distinction is made between the age groups "children" (defined age range: 1-12 years) and "adolescents" (defined age range: 13-23 years). "Children" are further subdivided into "toddlers" (defined age range: 1-3 years) and "older children"(defined age range: 4-12 years).”
  • 4, ll. 151-153;n=46 (65.7%) were children and n=24 (34.3%) were adolescents. When further subdivided, n=20 (28.6%) of the “children” were toddlers and n=26 (37.1%) were older children.”
  • 5, ll. 160-162;In both the “toddler” subgroup and the “older children” subgroup, exactly half of the patients had a sleep disorder (toddler: n=10; older children: n=13). In the "adolescents" subgroup, n=9 (37.5%) patients had a sleep disorder and n=15 (62.5%) did not”.
  • 5, ll. 167-174;In the subgroups, n=10 (50%) toddlers, n=11 (42.3%) older children, and n= 9 (37.5%) adolescents experienced “Chronic Insomnia”; n=8 (40%) toddlers, n=11 (42.3%) older children, and n=8 (33.3%) adolescents experienced a “Circadian Sleep-Wake Disorder”; n=1 (5%) toddler, n=4 (15.4 %) older children, and n=5 (20.8%) adolescents showed “Hypersomnia”, n=2 (10%) toddlers, n=2 (7.7 %) older children, and none of the adolescents experienced a “Sleep Related Movement Disorder”; n=1 (5%) toddler, n=2 (7.7 %) older children, and none of the adolescents showed “Parasomnia”. Chi-square tests revealed that there was no difference in prevalence of any sleep disorder across patient age subgroups (all p> .05).”

Reviewer 2 Report

Figure 1. is not clear enough, authors should add more results and use several statistical metrics to show the differences among sleep stages.

Please add related work section

Highlight the future work, and limitation of your study.  

Author Response

Insights into the frequency and distinguishing features of sleep disorders in pediatric palliative care incorporating a systematic sleep protocol

children-1049545

Figure 1. is not clear enough, authors should add more results and use several statistical metrics to show the differences among sleep stages.

Thank you very much for your helpful comments and the opportunity to revise our paper with regard to your suggestions. We have made various changes to Figure 1 and other parts of the manuscript to enhance the clarity of Figure 1:

  • As a prerequisite for understanding Figure 1, we have highlighted more clearly in Table 3 which characteristics depict pathological behavior typical of a specific sleep disorder: 6, ll. 192-194;The pathological behavior typical of a specific sleep disorder (derived from ICSD-3 criteria) is highlighted in bold letters.”
  • We have added a sentence that gives a clearer description of what can be seen in Figure 1: 6, ll. 201-202;Figure 1 shows the differences between patients with and without the different sleep disorders with respect to the relevant pathological behaviors (derived from ICSD-3 criteria):”
  • In Figure 1 itself, we have included more statistical characteristics/results and have included the names of the sleep disorders considered in a), b), and c): Please see Figure 1

Please add related work section

We have clarified the passages in which we refer to related work:

  • 8, ll. 233-234;The high prevalence rate of sleep disorders corresponds to related work […]”
  • 8, ll. 239-240;This finding is in line with results from related studies based on parental judgement [22,32-35].”
  • In addition, we have fixed a transfer error of the literature processing program, which caused references to be listed incorrectly in the bibliography.

Highlight the future work, and limitation of your study.  

In the "Limitations of the study" section of the Discussion, we have added two more aspects to further highlight the limitations of the study:

  • 9, ll. 296-301;Because some important other sleep disorders (Sleep-Related Breathing Disorders; Parasomnia and Sleep-Related Movement Disorders when identifying sleep-related characteristics) were not considered in this study, it is vital that this is taken into account when interpreting the study results and should not be misconstrued to mean that these non-focused sleep disorders are not present or not relevant in the sample studied.”).
  • 9, ll. 311-314; “Regarding the diagnosis of sleep disorders in this study, another important limitation must be mentioned: The judgment about the presence or absence of a corresponding sleep disorder was based only on the professionals' subjective judgment and was not supplemented by additional (objective) measures”.

Additionally, we have highlighted the discussed future work: pp. 9-10, ll. 326-329;Aspects to future interesting research have been mentioned above. In general, future research efforts should build on our study`s findings in order to continuously improve the quality of the diagnosis of sleep disorders in critically ill children and adolescents.”

Round 2

Reviewer 1 Report

The authors responded to the points I raised.

Author Response

Thank you very much for approving our revision and for your support.

Reviewer 2 Report

Authors have address my comments 

Author Response

(The authors gave the same response as above.)
